# Early Feasibility of an Activity-Based Intervention for Improving Ingestive Functions in Older Adults with Oropharyngeal Dysphagia

**DOI:** 10.3390/geriatrics8020044

**Published:** 2023-04-19

**Authors:** Tina Hansen, Louise Bolvig Laursen, Maria Swennergren Hansen

**Affiliations:** 1Physical Medicine & Rehabilitation Research—Copenhagen (PMR-C), Department of Occupational Therapy and Physiotherapy, Copenhagen University Hospital Amager and Hvidovre, 2650 Hvidovre, Denmark; 2Department of Orthopedic Surgery, Copenhagen University Hospital Amager and Hvidovre, 2650 Hvidovre, Denmark

**Keywords:** skill-based training, strength training, deglutition disorders, sarcopenia, intervention development, early-phase intervention, feasibility, acceptability

## Abstract

There is growing awareness about the use of combined strength- and skill-based swallowing training for improving swallowing physiology in the event of dysphagia. Such an approach involves focusing on coordination and timing as well as swallowing strengthening in the context of increased exercise complexity in eating and drinking activities. This study aimed to determine the early feasibility of a newly developed 12-week intervention, named the ACT-ING program (ACTivity-based strength and skill training of swallowing to improve INGestion), in older adults with dysphagia and generalized sarcopenia. In a multiple-case-study design, seven participants above 65 years of age (five women and two men) with slight to severe dysphagia and indications of sarcopenia underwent the intervention during hospitalization and in the community after discharge. The ACT-ING program met most of the feasibility marks in terms of demand (73.3% of those invited accepted participation), safety (100%), no reports of adverse events, tolerance (85.7%), usability (100%), and acceptability (100%). Three putative mediators of change (experienced autonomy support, in-therapy engagement, and perceived improvement in swallowing capacity) appeared to have been best accomplished in participants with slight to moderate dysphagia. The ACT-ING program showed preliminary evidence of early feasibility, warranting further early-phase dose articulation and proof-of-concept trials.

## 1. Introduction

Oropharyngeal dysphagia (OD) reduces the efficiency and safety of swallowing and is significantly more likely to develop in older adults (65+ years old) compared to the general population [1]. The condition is associated with severe complications, such as malnutrition, dehydration, aspiration pneumonia, increased hospitalizations, increased mortality, and an affected quality of life [2,3]. In recent years, OD in older adults has been related to sarcopenia as a whole-body process that also affects the masticatory and swallowing muscles (i.e., sarcopenic OD) [2,3,4]. The estimated prevalence of sarcopenic OD is around 30% in acute care hospitals and rehabilitation settings [5,6].

All stages of the complex swallowing process, involving the coordinated sensory–motor activity of the structures in the mouth, pharynx, larynx, and esophagus, may be altered by sarcopenia in the muscles of the upper aerodigestive tract [3,6]. A key intervention for managing sarcopenia is progressive resistance exercise [7]. Exercise modalities for improving oropharyngeal muscle strength, such as tongue-to-palate resistance training, the Shaker exercise, tongue-strengthening exercises using oral manometer devices, and Chin Tuck against resistance, can increase strength of the swallowing musculature [5,8]. However, when the intervention has ended, the muscle strength and the ingestion of foods and liquids tend to return to the pre-intervention level [9,10]. Since sarcopenia is not only attributed to a loss of muscle mass and strength but also to changes in complex neurological factors, such as hypo-excitability in both upper and lower motor neurons, it is suggested to incorporate goal-based motor skill training into the strength training [7]. Combined strength- and skill-based training for improving swallowing physiology involves focusing on coordination and timing as well as the progressive strengthening of swallowing in the context of increased exercise complexity in eating and drinking tasks and activities [11,12]. The underlying premise is to enhance experience-dependent neuroplasticity and induce neuromuscular excitability which, in turn, facilitates the improvement of task performance [7].

Recently, Carnaby et al. [13] found promising results in terms of a reduction in OD severity, improved oral intake, and earlier return to pre-intervention diet in a population of stroke patients who participated in a combined strength- and skill-based training approach. Inspired by this approach [14], Hansen et al. [15] developed the ACT-ING program (ACTivity-based strength- and skill-based training of swallowing to improve INGestion) for older adults at risk of or with sarcopenic OD. The intervention development included several literature reviews and the involvement of stakeholders and experts [15]. This resulted in an intervention manual structured according to the “Template for Intervention Description and Replication” (TIDieR) which covers 12 items reflecting all necessary elements for reporting non-pharmacological interventions [16]. The description of the ACT-ING program is also presented in the Appendix A. The program theory is based on a task-oriented approach [17,18] combining strength- and skill-based training principles [11,12] in a person-centered climate [19], which is guided by a theory of human motivation (self-determination theory) [20]. The fundamental proposition of the ACT-ING program is that when the participant feels that their basic psychological needs (i.e., feelings of autonomy, competency, and relatedness) have been satisfied, it will promote autonomous motivation for engaging in the therapy and contribute to a sense of swallowing capacity. Accomplishing the intervention’s putative mediators of change is expected to be essential for acquiring positive swallowing-related short-, medium-, and long-term outcomes [21]. This is presented in Appendix A.

The design and evaluation of health interventions is an iterative, multi-phase process [22,23,24]. Currently, the ACT-ING program is an “early-stage” intervention with a low level of maturity, necessitating early feasibility testing in a real-world setting [24]. Therefore, the aim of this study was to determine the early feasibility of the ACT-ING program with a specific focus on functionality (i.e., whether the intervention can work as intended) and including usability and acceptability from the perspective of the participants. In addition, the achievement of the putative mediators of change was explored, as these mediators constitute the intermediate processes [21] through which the ACT-ING program is expected to achieve its effect on swallowing-related outcomes. The research questions were:(1)Does the ACT-ING program meet a set of a priori feasibility marks?(2)Is the ACT-ING program perceived as usable and acceptable by older adults with OD who are participating in the intervention?(3)Have the putative mediators of change of the ACT-ING program been accomplished?

## 2. Materials and Methods

### 2.1. Design

The study used a mixed-methods, multiple-case-study design [25] with quantitative and qualitative data used in a convergent approach [26]. This design is useful for exploring a real-life contemporary bounded system (a case) or multiple bounded systems (cases) over time through multiple sources of information [25,26]. According to Yin [25], the unit of data collection is the source of evidence in a case study, while the unit of analysis is the case itself. This study included multiple participants with OD who were engaged in the ACT-ING program. In this study, procedures were replicated for each case, which represented individual units of analysis [25]. Patterns of similarity or difference were explored [26] to identify the early feasibility of the ACT-ING program.

### 2.2. Setting

The study took place at an acute hospital in the Capital Region of Denmark and recruited participants transferred to internal medical wards for further observation and treatment after acute admission to the emergency department. Rehabilitation efforts may begin in the inpatient setting, where the average length of stay in Denmark is about 3.5 days [27]. When a patient needs OD management during hospitalization, the wards send an electronic referral for an occupational therapist (OT) from a central physiotherapy and occupational therapy department. At the time of the study, OD management was delivered by three OTs. Upon discharge, an individual rehabilitation plan describing the patient’s current functional level and rehabilitation needs was sent to the home municipality, and the patient was contacted by a municipal OT. For this study, two municipalities from the uptake area of the hospital participated. Three OTs were identified by the local manager of the municipal rehabilitation services as they had a special interest in OD management and related post-graduate education and experience. Note that the number of participating municipal OTs was constrained by prevention actions in terms of social distancing and minimizing social contact during the COVID-19 pandemic in 2020–2022.

### 2.3. Participants

Patients who were above 65 years of age and were referred for OD management during hospitalization as inpatients or outpatients were included according to the following criteria: the participants were residents of the participating municipalities; spoke and understood Danish; were able to provide written informed consent; were sufficiently awake (a minimum of 15 min), alert, and oriented; were able to participate in a short oral examination (open mouth, stick out tongue, and smile); and experienced suspected OD, which was assessed by the Gugging Swallowing Screen (GUSS) [28]. The GUSS includes four swallowing subtests: (1) vigilance and saliva swallowing and the ingestion of (2) semisolids, (3) liquid, and (4) solid foods. These subtests are evaluated based on points, with a maximum of five points in each subtest and a total score of 20 indicating normal swallowing function. Patients were excluded if they had evidence of OD related to the esophagus, a psychiatric diagnosis, a neurodegenerative disease, cognitive dysfunction, or required palliative care.

A sample size is not calculated per se for case studies [25]. To capture the early feasibility of the ACT-ING program in rich detail, we aimed to recruit five participants within each of three OD severity groups according to the GUSS: severe OD (0–9 points); moderate OD (10–14 points); and slight OD (15–19 points) [28].

### 2.4. Intervention

The ACT-ING program functions as an adjunct to the usual management of OD. In the current study, this included determining the efficiency and safety of eating and swallowing, adjustments in sitting position, and diet recommendations.

The ACT-ING program is described according to TIDieR [16] in the Appendix A. In short, the ACT-ING program aims to maximize swallowing-related outcomes and prevent further decline of the swallowing muscles in older adults at risk of or with sarcopenic OD [15]. The dosage is 2–3 individual, face-to-face therapy sessions per week for up to a maximum of 12 weeks. A therapy session lasts up to 45 min. In between therapy sessions, participants perform self-training during daily meals and document this in a food diary [15]. The intervention can be finalized before the end of the 12 weeks if participants achieve their goal or reach a sufficient level of functional oral intake (i.e., a total oral diet of multiple consistencies without special preparation but with a few specific avoidances or limitations [29]). The ACT-ING program uses goal-directed and task-specific swallowing exercises in eating and drinking activities to improve the strength, speed, and coordination of swallowing [15]. Initially, participants are taught a swallowing strategy (effortful swallowing) to voluntarily increase the movement of the oropharyngeal structures and the generation of pressure in the muscles of the tongue and the pharynx [30]. When the participant masters the swallowing strategy, advancing steps of an altered bolus volume, bolus consistency [31], and swallowing repetitions are introduced according to a 17-level task hierarchy and predetermined progression rules (Appendix A).

Since this was the very first time the ACT-ING program was tested, the developer and principal investigator T.H., who is affiliated with the hospital setting, served as a consistent interventionist in close cooperation with the participants’ treating OTs in both settings. For the in-therapy sessions, T.H. attended the participants’ location at the hospital and/or in the community (own home or rehabilitation center). Due to COVID-19-prevention actions, it was not possible to carry out a planned 8 h training course for the participating OTs. Instead, they received the intervention manual for preparation, were provided with a 1 h individual introduction, observed therapy sessions delivered by T.H., and were responsible for delivering therapy sessions when T.H. was unavailable. If required, the OTs were supervised by T.H. to ensure consistency in the delivery of the intervention. In the very early phase of the study, feedback from the OTs resulted in adjustments to a few intervention components but without alterations to the major intervention principles and techniques (Appendix A, TIDieR items 3, 4, 6, and 10).

### 2.5. Data Collection

Data were collected at baseline, during the intervention, and post intervention.

#### 2.5.1. Baseline Assessment

Baseline assessments were carried out to obtain a profile of each participant. In addition to the information on the severity of OD obtained using the GUSS, information on age, gender, admission diagnosis, comorbidity severity, indication of sarcopenia of the whole body and of the swallowing muscles, nutritional status, mealtime performance, and functional oral intake was included.

Comorbidity severity was obtained using the age-adjusted Charlson Comorbidity Index (aCCI) [32].

An indication of sarcopenia of the whole body was determined using the SARC-F [33] and cut-off values for handgrip strength (HGS), as recommended by the European Working Group on Sarcopenia in Older People [34]. The SARC-F is a 5-item questionnaire (strength, assistance with walking, rising from a chair, climbing stairs, and falls) rated on a 3-point ordinal scale from 0 (no difficulty) to 2 (significant difficulty). A total score of ≥4 out of 10 points is indicative of a risk of sarcopenia [33]. HGS was measured in kilograms (kg) using a Jamar hydraulic hand dynamometer, and the maximum of three consecutive measurement trials of the dominant hand was used [35]. The cut-off points to indicate probable sarcopenia were <27 kg for men and <16 kg for women [34].

An indication of sarcopenia of the swallowing musculature was determined from the maximum tongue pressure (MTP), using the Iowa Oral Performance Instrument (IOPI) [36]. The highest pressure across three trials in kilopascals (kPa) was used. Reduced tongue strength was reflected by an MTP score < 34 kPa, which is below the 5th percentile from the estimated normal distribution for older adults across both genders (60+ years of age) [37]. The cut-off value indicative of sarcopenia of the swallowing musculature was MTP < 20 kPa [4].

Nutritional status was assessed with the Mini Nutritional Assessment Short Form (MNA-SF) [38], which includes six items related to food intake, unintentional weight loss, neuropsychological problems, acute disease, mobility, and anthropometric measurements (body mass index). The maximum possible score is 14 points, and the nutritional status is determined as malnutrition (MNA-SF < 8), risk of malnutrition (MNA-SF = 8–11), or no malnutrition (MNA-SF = 12–14).

Mealtime performance was assessed with the McGill Ingestive Skills Assessment, Version 2 (MISA2) [39], an observation-based clinical assessment of a test meal. The MISA2 consists of 36 ingestive skill items scored on a 3-point ordinal scale from 1 (absent ingestive skill) to 3 (adequate ingestive skill performance) which are summated into a total score ranging from 36 to 108. Lower scores indicate impaired mealtime performance.

Functional oral intake was determined using the Functional Oral Intake Scale (FOIS), which covers seven levels of varying degrees of oral feeding ranging from tube-dependent (level 1) to a regular diet (level 7) [29]. The FOIS was rated based on the results obtained with the MISA2 [39].

#### 2.5.2. Feasibility Marks

Six feasibility marks were decided a priori [22,23,24,40] as follows:The demand for intervention by the target group was assessed as the proportion of eligible participants who were invited and agreed to participate. The success criterion was a proportion of ≥70%.Retention was assessed as the proportion of enrolled participants who completed the post-intervention assessments. The success criterion was a proportion of ≥85%.Intervention adherence was assessed as the proportion of enrolled participants who attended at least 75% of the planned therapy sessions. Adherence to self-training was assessed as the proportion of enrolled participants who completed at least 75% of their weekly food diary. The success criterion for both was a proportion of ≥70%.Safety was assessed during each therapy session using records on clinical signs of aspiration (e.g., wet voice, throat clearing, coughing, or gagging) which might increase briefly during exercise progression but are expected to decrease as the participants’ skills increase [41]. The success criterion was that clinical signs of aspiration occurred in less than 20% of the therapeutic swallowing attempts in 80% of the therapy sessions for 100% of the participants.Adverse events were assessed using records of any unexpected and unintended serious events related to the ingestion of training material during the therapy sessions (e.g., food allergy symptoms, severe pain, choking, and apnea). The success criterion was no record of adverse events.Tolerance was assessed at the end of each therapy session, when participants rated their experienced level of concern for aspiration on a 100 mm visual analog scale (VAS) with a horizontal line (left side = not concerned at all (0 mm); right side = extremely concerned (100 mm)). The distance from the left edge of the line to the mark placed by the participant was measured to the nearest millimeter and used in the analyses. The success criterion was that 80% of the aspiration concern VASs were ≤70 mm for at least 85% of participants.

#### 2.5.3. Usability and Acceptability

Intervention usability was assessed post intervention by the Intrinsic Motivation Inventory (IMI) "Value/usefulness” subscale with seven items addressing the content and level of motivation that a participant experiences during an intervention [42]. Items are scored on a seven-point Likert scale ranging from 1 (not at all true) to 7 (very true). A neutral score on the IMI is four (somewhat true), with a higher score indicative of a more positive result for motivation. The success criterion was that the “Value/usefulness” subscale score was >4 (average score across 7 items) for 100% of the participants.Acceptability was assessed during each intervention using fieldnote records of participants’ reactions and post intervention by a series of evaluation questions with a blend of closed- and open-ended questions based on the Theoretical Framework of Acceptability (TFA), which covers seven dimensions of intervention acceptability: (1) affective attitude (how the participant feels about it), (2) burden (perceived amount of effort required to participate), (3) ethicality (whether it fits with the participant’s value system), (4) intervention coherence (whether the participant understands it and how it works), (5) opportunity costs (whether benefits, profits, or values must be given up for participation), (6) perceived effectiveness (whether it is perceived as likely to achieve its purpose), and (7) self-efficacy (whether the participant has confidence in his/her own ability to perform the actions required to participate) [43]. The criterion was that the participants’ responses reflected that the intervention was acceptable.

#### 2.5.4. Putative Mediators of Change

The satisfaction of basic psychological needs was assessed post intervention by the Basic Psychological Needs in Exercise Scale (BPNES) [44], a participant-reported questionnaire concerning the extent to which the innate psychological need for autonomy (4 items), competence (4 items), and relatedness (4 items) are satisfied in the intervention. Items are rated on a five-point Likert scale ranging from 1 (do not agree at all) to 5 (completely agree), with higher scores indicating a high degree of satisfaction of basic needs. It was expected that the scores for the three dimensions were > 3 (the average score across four items for each dimension).In-therapy engagement was assessed during each therapy session using records on accompanying worksheets for the key features of the intervention in terms of external exercise loads reflecting practice complexity (task hierarchy levels exercised), practice variability (number of task hierarchy levels per session), practice distribution (number of sets per session), and practice amount (number of swallows across sets and sessions). The internal exercise load after each set was also obtained by the OMNI Perceived Exertion Scale for Resistance Exercise (OMNI-RES), ranging from 0 (extremely easy) to 10 (extremely hard) [45]. It was expected that all key features were implemented across the intervention.The perceived swallowing capacity when ingesting liquids and foods was assessed at baseline and post-intervention using a 100 mm VAS scale (left side (0) = unable to swallow; right side (100 mm) = no difficulties). The distance from the left edge of the line to the mark placed by the participant was measured to the nearest millimeter and used in the analyses. It was expected that the participants perceived their swallowing capacity to have improved.

#### 2.5.5. Procedure

Data collection at baseline and post intervention was undertaken by T.H., and data collection during therapy sessions was performed by the participating OTs or T.H. The IMI “Value/usefulness” subscale, the BPNES, and the acceptability questions were asked in a face-to-face interview post intervention in the participant’s own home. The answers to the open-ended acceptability questions were written as brief notes during the interview and were subsequently written up in detail shortly after each interview. To minimize the desirability bias [26], the participants were told that it was very important for us to learn about any aspects of the intervention that needed optimization and that there were no right or wrong answers.

#### 2.5.6. Data Analysis

Descriptive statistics and qualitative approaches were used to explore the data. For the feasibility marks and putative mediators of change, frequencies and proportions were used. For the IMI “Value/usefulness” subscale and the three dimensions of the BPNES, proportions of participants with average scores above the neutral score were used. The fieldnotes and interview scripts regarding acceptability were prepared for analysis using Microsoft Excel and Word [46], and they were analyzed using a deductive content analysis approach applying the seven TFA domains and definitions as an a priori analysis framework template [26]. Although coding was primarily deductive, there was flexibility in creating subthemes within each of the TFA domains. Coding was undertaken independently by T.H. and L.B.L., and discrepancies were discussed and resolved.

The results guided the decision as to whether the ACT-ING program needs improvement. Three scenarios were formulated: (1) major modifications are needed because none of the feasibility marks were met and negative signals were observed for usability, acceptability, and the putative mediators of change, or adverse events were reported; (2) minor modifications are needed since some feasibility marks were not met but positive signals were observed for usability, acceptability, and the putative mediators of change; or (3) modifications are not needed since all feasibility marks were met, and there were positive signals for usability, acceptability, and the putative mediators of change.

## 3. Results

### 3.1. Demand for the Intervention and Retention

Recruitment ran from mid-September 2020 until October 2022 but was discontinued for 42 weeks due to the COVID-19 pandemic or conditions and absence of trial staff, resulting in an actual recruitment period of 62 weeks (15.5 months). A total of 475 patients (inpatients (*n* = 473) and outpatients (*n* = 2)) were referred for OD assessment, and 15 participants were identified as eligible and invited to participate (Figure 1). Eleven participants (73.3%) were enrolled in the intervention. Of those, four participants with severe OD died before initiation of the baseline assessments. Over the course of the intervention, two participants (cases 1 and 7) died due to advancing disease, and the cognitive functions of one participant (case 5) deteriorated, resulting in the premature termination of the intervention. Accordingly, data on the feasibility marks and the putative mediators of change during therapy were available for seven participants (63.6%), and post-intervention assessments of feasibility, usability, acceptability, and the putative mediators of change were available for four participants (36.4%).

### 3.2. Baseline Characteristics and Assessments

The characteristics and baseline assessments of the participants undergoing the intervention are presented in Table 1. All participants had experienced some degree of OD for several years. Based on the GUSS scores, OD severity was slight in cases 1, 2, 8, and 10, moderate in case 3, and severe in cases 5 and 7. The SARC-F scores indicated a sarcopenia risk for all participants, and the HGS indicated probable sarcopenia in cases 3, 7, and 8. Tongue strength was below the age norm for all but case 10 and below the cut-off value for sarcopenia in cases 5 and 7. The MNA-SF scores indicated that case 8 had a normal nutritional status, cases 2, 3, and 9 were at risk of malnutrition, and cases 5 and 7 were malnutritioned. Mealtime performance (MISA2) and functional oral intake (FOIS) were impaired for all participants.

Table 2 presents the most impaired ingestive skills observed during the MISA2 test meal for each case. Across all cases, the observed ingestive skill difficulties were predominantly related to mastication (i.e., ingestive skill items “uses functional chewing pattern” and “brings bolus into a cohesive unit”), coordination between swallowing and breathing (i.e., ingestive skill item “maintains respiratory pattern”), swallowing efficiency (i.e., ingestive skill item “swallows without extra effort”), and airway protection (i.e., ingestive skill item “protects the airway from penetration/aspiration”). Cases 1, 5, and 7 demonstrated ineffective throat clearing when there were episodes of observable signs of aspiration (i.e., ingestive skill item “coughs or clears the airway efficiently if needed”).

When setting goals for participation in the ACT-ING program, all patients emphasized ability in mealtime participation with access to preferred or familiar foods as well as dining with family or friends.

### 3.3. Intervention Adherence

Table 3 displays the overall intervention duration and weekly attendance for each participant. The intervention flow of at least two planned therapy sessions per week was discouraged for six participants, mainly due to COVID-19-prevention actions, participant condition, or scheduling challenges. Across cases, the average (SD) number of attended therapy sessions was 15.6 (6.0). Overall, four out of seven participants (57%) attended ≥75% of the planned therapy sessions, which lasted about 20 to 45 min. Case 5 underwent three therapy sessions per week due to inability to perform self-training independently. Of the six participants able to perform self-training, three completed the weekly food diary only once and three did not complete it at all.

### 3.4. Safety, Adverse Events, and Tolerance

No adverse events during the therapy sessions and self-training were recorded. Table 4 shows that clinical signs of aspiration occurred in less than 20% of the therapeutic swallowing attempts for more than 80% of the therapy sessions in all participants, indicating that the criterion for the feasibility mark of safety was met. The criterion for the feasibility mark of tolerance (i.e., 80% of the VAS scales ≤ 70 mm) was met in six out of seven participants (85.7%). For case 5, the criterion was only met in about 71% of the therapy sessions. However, missing values were present in cases 1, 2, 3, and 7 due to incomplete data collection.

### 3.5. Intervention Usability and Acceptability

Table 5 presents a summary of the findings for IMI “Value/usefulness” subscale and the acceptability questions based on the eight TFA domains for the four participants who completed the post-intervention assessments.

The IMI “Value/usefulness” subscale score was >4 for the four participants, indicating that the success criterion for usability was met.

Regarding intervention acceptability, the participants expressed a positive affective attitude towards the ACT-ING program, which was related to a sense of pleasure (cases 2, 3, 8, and 10) and security (case 2). The burden in terms of duration, frequency, and session length was evaluated as appropriate by the participants, and cases 3 and 10 reported that the consecutive visits helped to minimize failure in performance. However, the food diary was experienced as a burdensome component (cases 2, 3, and 8). Concerning the program’s fit with personal values (ethicality), the opportunity for support was emphasized by the participants. In relation to intervention coherence, the participants articulated an understanding of the basic assumption of the ACT-ING program in terms of volitional swallowing and awareness during ingestion (cases 2, 3, and 10) and that the training material assisted in skill acquisition (cases 8 and 10). None of the participants reported any opportunity costs of engaging in the intervention, and the participants perceived a flexible schedule (cases 3, 8, and 10). The participants experienced improvements in their ingestive skills (perceived effectiveness) and expressed a sense of capacity and confidence in their ability to perform the effortful swallow during ingestion (self-efficacy). When asked how the intervention could be improved, the participants expressed satisfaction with the current intervention content.

### 3.6. Putative Mediators of Change

Table 6 displays the descriptives of the putative mediators of change. The four participants completing the post-intervention assessments rated their satisfaction of the three basic psychological needs in the exercise as high. In-therapy engagement is reflected for all seven participants with an increase in external training loads during therapy. For participants with slight or moderate OD (cases 1, 2, 3, 8, and 10), the practice complexity, variability, distribution, and amount were, however, more noticeable than for the two participants with severe OD (cases 5 and 7). Figure 2 shows that for most participants, the internal training load delivered across all sets predominantly ranged within the areas of somewhat easy and somewhat hard. Post intervention, the four participants completing the intervention showed improved perceived swallowing capacity and expressed that their goal for participating in the ACT-ING program was achieved.

## 4. Discussion

This study is part of a large iterative intervention development process, and it focused on the early feasibility of the ACT-ING program, which uses goal-directed and task-specific swallowing-strengthening exercises in eating and drinking activities. To a large extent, focusing on the early feasibility of a new intervention aligns with the procedures that are known, accepted, and expected for drug development [23]. We found that most of the feasibility marks for functionality, usability, and acceptability appeared satisfactory, and that the putative mediators of change in the ACT-ING program seemed to be accomplished.

Although the overall recruitment rate was very low, the demand for the intervention was acceptable given that 73% of the 15 patients successfully contacted agreed to participate. However, the retention rate was lower than anticipated, with only about 36% completing the post-intervention assessment. The reason for this high attrition rate was predominantly a result of death due to advancing disease. It is well-known that longer consent and recruitment procedures and high attrition rates due to death are barriers within geriatric research [47]. For future trials, it might be appealing to exclude individuals with several comorbidities to maximize the chances of success. However, since the ACT-ING program is tailored to vulnerable older adults [15], this would be inappropriate. Instead, alternative trial designs, such as randomized, single-subject experimental designs [48], could be suggested for future preliminary and efficacy testing.

The intervention adherence rates ranged from 56% to 100%, and only four of the seven participants (57%) met the criterion of an attendance rate of the planned therapy >75%. The reasons for non-adherence were advanced disease or rehospitalization, COVID-19-prevention actions, and general scheduling problems. Within geriatric research, low intervention adherence to exercise interventions is an issue and is associated with the severity of symptoms, multimorbidity, and frailty [49]. It is suggested that intervention adherence is a concept with deeper roots in the participant’s behavior than the percentage of sessions attended and that the degree to which the target intensity and volume are achieved might be more important [49]. In the current study, the key features of the intervention in terms of increased practice amount, distribution, and variability could be accomplished. As such, the participants’ intervention adherence might be interpreted as acceptable. In addition, the average number of attended therapy sessions was about 16 across participants. This is somewhat higher than in the study by Carnaby et al. [13], who planned daily therapy sessions over a consecutive 3-week period and achieved an average number of attended therapy sessions of about nine. Thus, it might be speculated whether our criterion for this feasibility mark was too high. In fact, the participants who completed the intervention perceived the duration and frequency as appropriate and not too demanding, and they valued the flexibility in scheduling. However, since some non-adherence was due to scheduling problems, few sessions could be substituted by telehealth consultation, which has emerged within OD management as a cost-effective mode of delivery [50]. Having said this, resources in the health care system are limited [27], and long-duration interventions, such as the ACT-ING program, are costly [49]. In addition, it is known that the longer the duration of an intervention, the lower the adherence obtained in older participants [49]. This suggest that in addition to being individually tailored according to the level of the participants’ skills, the ACT-ING program should also emphasize individualization of duration and frequency for effective promotion of adherence [49]. Therefore, there is a need for further research on the most optimal intervention dose (i.e., duration, number of sessions, session length and density, and task length, difficulty, and intensity) within the resources available and how optimal dosing is influenced by factors such as age, sex, comorbidities, primary etiology, or physical fitness [10,51].

It is worth noting that the intention of recruiting five participants within each OD-severity group was not realized in the current study. Of the 11 participants initially enrolled, 6 had severe OD, and none completed the intervention. In addition, the accomplishment of the key features of the intervention was less notable for two participants (cases 5 and 7) who had severe OD and an indication of sarcopenia of the swallowing musculature. Accordingly, it might be speculated whether the ACT-ING program is best suited to participants with slight to moderate OD and less pronounced indications of sarcopenia.

The safety criterion of clinical signs of aspiration in less than 20% of the therapeutic swallowing attempts was achieved for all the participants. In addition, adverse events were not reported, which aligns with similar intervention approaches [41]. However, the safety criterion was only just met for cases 5 and 7, which may further signal that the ACT-ING program might be less suitable for participants with severe OD. Furthermore, the intervention was well-tolerated by participants except for case 5, who had severe OD and deteriorated in cognitive function. In the ACT-ING program, exercise progression/reversion is based on perceived exercise exertion, and clinical signs of aspiration are only accepted in less than 20% of the therapeutic swallowing attempts after advancing to a more difficult task level [15]. This is stricter than the approach in Carnaby et al. [13] in which up episodes of clinical signs of aspiration of up to 40% are accepted before reversion [14]. Since enjoyment and feeling safe without aspiration during ingestion and mealtimes are perceived as essential in older adults with OD [52], the strict criterion in the ACT-ING program might have contributed to the intervention tolerance, perceived usability, and acceptability.

Intervention usability and acceptability among the participants was reflected by positive motivation scores on the IMI subscale and by a positive affective attitude toward the intervention content and components, except for the food diary for documenting self-training, which was perceived as burdensome and was omitted. This might reflect that the food diary might have functioned as a type of external regulation of motivation [20]. This is not ineffective per se, but without clear expectancy, this might result in a lack of maintenance and sustained behavior [20]. In fact, the food diary aims to monitor adherence to self-training rather than represent personal value for the participant. This is not in accordance with the intention of the person-centered perspective in the ACT-ING program, in which the participants’ autonomous motivation for participation is proposed to be an important mediator [15]. In the current study, the food diary was substituted by conversations during the therapy sessions, which was found to be more acceptable to the participants. For future research, including a limited checklist of foods and beverages customized to each participant’s oral functional level and with a frequency response section on how often each item is consumed between therapy sessions could be suggested. This will be relatively inexpensive, place less of a burden on the participant, and could contribute to reflections on the degree to which participants apply the therapy in their daily meals during the intervention period. In this way, the integrated regulation of motivation, the most autonomous form of extrinsic motivation [20], could be supported.

The putative mediators of change in terms of satisfied basic psychological needs, in-therapy engagement, and improved perceived swallowing capacity seemed to be accomplished. In particular, it appears that relatedness satisfaction was rated very highly across the participants who completed the intervention, which might signal that the participants felt accepted and respected and that they valued the feeling of emotional and social connection to the OTs. However, we were not able to verify the pathway of the putative mediators. For such understanding, sophisticated analyses with a path analysis or structural equation modeling [53] are needed. Regarding the in-therapy engagement, there was an unexpected result in relation to the internal training loads. These were predominantly perceived to be within the areas of somewhat easy/somewhat hard, which corresponds to the training zones for muscular endurance and power [45]. In the current study, the participants were instructed to use effortful swallowing, which has been shown to increase the tongue-to-palate maximum pressure generation during the initial stage of swallowing [30]. From the perspective of the participants, effortful swallowing became a routine. This might explain why the participants only perceived exercise exertion as hard for a relatively small percentage of the training sets. In addition, swallowing is a submaximal task [11,12], and translating training zones for muscular strength, endurance, and power for the resistance training of skeletal muscles to swallowing might not be appropriate. There are differences in muscle composition, sensorimotor complexity, and neural processes [11,12]. However, using the OMNI-RES during the therapy sessions might have facilitated the accomplishment of the practice variability, distribution, and amount.

### Methodological Considerations

Some methodological aspects need to be considered. The study was performed during the COVID-19 pandemic, which restricted the participant recruitment, the intervention flow, and the involvement of several interventionists as well as their training. The consequences included a very small sample and low intervention adherence. In addition, the intervention was delivered by the developer and carefully selected OTs who were supervised by the developer. Thus, there is a high risk of delivery agent bias and implementation support bias [40]. There might also be a high risk of observer expectation bias for the data collection during the therapy session as well as a high risk of reporting and interviewer bias for the data collected post intervention. Such biases can overestimate the feasibility and underestimate the complexity of the intervention [40]. However, the setup allowed for the prompt adjustment of a few intervention components during the current phase of the intervention development.

In current study, suspected OD was assessed using the GUSS [28], and further clinical swallow assessments included tongue strength assessments using the IOPI [36], ingestive skills during a test meal, assessed using the MISA2 [39], and functional oral intake, assessed using the FOIS [29]. However, the OD assessment process should optimally involve further instrumental assessment (i.e., fiberoptic endoscopic evaluation of swallowing or videofluoroscopic evaluation of swallowing), which are considered the gold standard for detecting aspiration (including silent aspiration) [54]. In addition, the evaluation of the feasibility mark for safety was based on the observation of clinical signs of aspiration. However, silent aspiration might have occurred during the therapy sessions.

An indication of sarcopenia of the whole body was determined using SARC-F and HGS, and sarcopenia of the swallowing musculature was determined using MTP. However, to definitely confirm sarcopenia and sarcopenic OD, measures of muscle quantity (e.g., appendicular skeletal muscle mass) and quality (e.g., micro- and macroscopic changes in muscle architecture and composition) are needed [4,34]. In addition, SARC-F has been criticized for its inability to detect mild cases due to low-to-moderate sensitivity [34].

It is also worth noting that the IMI “Value/usefulness” subscale and BPNES have not been validated in a Danish context. In addition, the VAS for an aspiration concern, which was used for investigating the feasibility mark tolerance, was developed for this study instead of using the VAS for anxiety (VAS-A) [55]. The reason for this choice was that the concern is specifically related to a concrete situation (e.g., concerns about aspiration during therapy) and is often temporary, whereas anxiety is more generalized and longstanding [55]. The determined upper limit threshold of 70 mm was based on the literature on VAS for pain [56] and for anxiety [57], in which ratings ≥ 70 mm were reported as a valid cut-off for severe levels of the measured construct [56,57]. However, this threshold might not be a valid indicator for aspiration concern (i.e., intervention tolerance), and a lower cut-off could have been chosen. Nevertheless, the perceived usability and acceptability of the intervention were positive, which might indicate adequate intervention tolerance.

Finally, it could be argued that post-intervention outcome data should have been added. However, our study did not aim to investigate the preliminary efficacy of the ACT-ING program but instead focused on the early feasibility as well as the putative mediators of change. This priority is consistent with recommendations for the development of early-stage interventions [20,21,22,40]. In addition, further testing of feasibility and fidelity will be needed before continuing on to large-scale efficacy trials.

## 5. Conclusions

In conclusion, the results from this early feasibility study suggest that the 12-week ACT-ING program seems to be safe, well-tolerated, usable, and acceptable in older adults with slight to moderate OD. In addition, the putative mediators of change of the intervention in terms of satisfied basic psychological needs, in-therapy engagement, and improved perceived swallowing capacity were shown to be accomplished. However, the intervention might benefit from minor adaptations related to the duration and the self-training between therapy sessions, which need to be addressed in further early-phase dose articulation and proof-of-concept trials. In addition, recruitment and retention within the hospital setting was difficult. This has consequences for the design of subsequent trials in the multi-stage iterative process of the development and evaluation of the ACT-ING program.

## Figures and Tables

**Figure 1 geriatrics-08-00044-f001:**
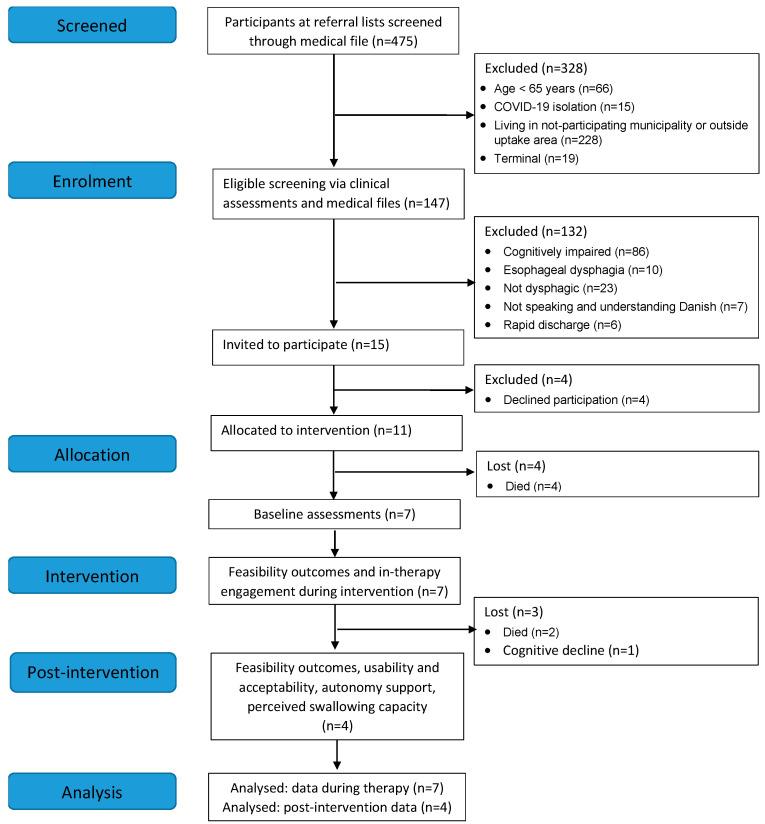
Flow diagram for study participants.

**Figure 2 geriatrics-08-00044-f002:**
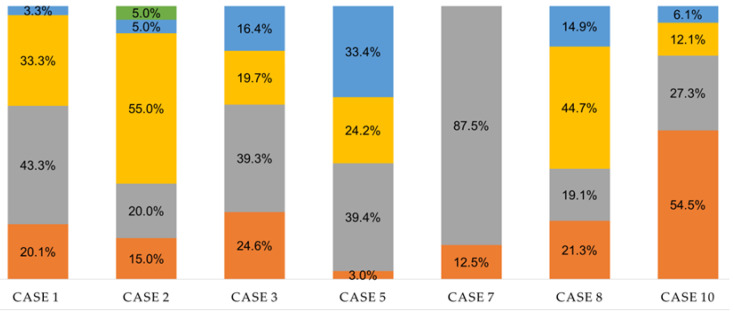
Percentage distribution of internal exercise loads across all sets. Perceived exertion assessed with a rating of perceived exertion (RPE) scale from 0 (extremely easy) to 10 (extremely hard) [45]. The color codes are: green—extremely hard (9–10); blue—hard (7–8); yellow—somewhat hard (5–6); grey—somewhat easy (3–4); orange—easy (1–2). Extremely easy (0) was not reported.

**Table 1 geriatrics-08-00044-t001:** Baseline demographic and clinical characteristics.

Case	Age	Gender	Admission Diagnosis	aCCI	GUSS	SARC-F	HGS	MTP	MNA-SF	MISA2	FOIS
1	86	Male	Pneumonia	7	18	7	20	20	4	82	5
2	87	Female	Duodenal ulcer	7	18	8	20	28	8	73	4
3	84	Female	Aspiration pneumonia	5	13	9	12.5	20	8	77	4
5	85	Female	Diabetes mellitus	7	3	8	N/A	16	6	42	1
7	86	Male	Dehydration	6	9	6	19.5	13	6	71	3
8	78	Female	HNC sequelae ^#^	6	19	5	18.5	33	12	86	5
10	67	Female	HNC sequelae ^#^	5	19	7	25	36	9	88	5

Abbreviations: aCCI—adjusted Charlson Comorbidity Index [32]; FOIS—Functional Oral Intake Scale [29]; GUSS—Gugging Swallowing Screen [28]; HGS—hand grip strength, measured in kg; HNC—head and neck cancer; MISA—McGill Ingestive Skills Assessment [39]; MNA-SF—Mini Nutritional Assessment Short Form [38]; MTP—maximal tongue pressure, measured in kPa; SARC-F—Strength—assistance with walking—rising from a chair—climbing stairs—and falls questionnaire [33]. ^#^ HNC sequalae: case 8, surgery/radiotherapy after malignant tumors of the salivary glands two years ago; case 10, radiotherapy after epithelial tumors of hypopharynx 15 years ago.

**Table 2 geriatrics-08-00044-t002:** Impaired ingestive skills observed during the MISA2 test meal.

	Case						
Ingestive Skills	1	2	3	5	7	8	10
Seals lips on cup/glass/utensil			x	x			
Controls liquid bolus in mouth before swallowing			x	x			
Uses functional chewing pattern	x		x	x	x	x	x
Controls solid bolus in mouth before swallowing				x			
Brings bolus into a cohesive unit	x		x	x	x	x	
Transport bolus backwards in mouth			x	x	x	x	
Swallows without extra effort	x	x	x	x	x	x	x
Swallows only once or twice	x		x	x	x		
Maintains respiratory pattern	x	x	x	x	x		x
Protects the airway from penetration/aspiration	x	x	x	x	x	x	x
Coughs or clears the airway efficiently if needed	x			x	x		

MISA—McGill Ingestive Skills Assessment [39].

**Table 3 geriatrics-08-00044-t003:** Delivery of the intervention to participants by week.

Week	1	2	3	4	5	6	7	8	9	10	11	12	Attended/Planned (Adherence)
Case 1	IEE	EEE	EE	EE	CC	CC	EE	EC	EE	T			15/20 (75%)
Case 2	IEE	EE	CC	EC	EC	CC	CC	EE	EE	EF			13/21 (62%)
Case 3	IEE	EE	EE	EE	EC	EE	EE	EE	EE	EC	CC	F	20/24 (83%)
Case 5	I	IEE	EEE	EEE	EEE	EEE	EEE	ECC	EEE	EEE	T		26/28 (93%)
Case 7	IEE	EE	EE	EE	EE	T							11/11 (100%)
Case 8	I	EE	EE	CC	EC	CC	EE	EC	EC	EC	EC	F	13/22 (59%)
Case 10	I	CC	IE	CC	EE	EC	EC	EC	CC	EC	F		10/20 (50%)

Labels: I—initial session (informed consent, assessments); E—exposure session (face to face therapy); C—circumstances of participant’s situation or environment that discourage therapy flow (COVID-19 restrictions, rehospitalization, and scheduling challenges); F—final session (post-intervention feasibility outcomes); T—termination procedure due to participant condition. More than one label within a week cell indicates multiple sessions in the same week. Colors: gray cell indicates rehabilitation in hospital; white cell indicates community-based rehabilitation.

**Table 4 geriatrics-08-00044-t004:** Feasibility marks: safety and tolerance during intervention.

Case ID	Safety Clinical Signs of Aspiration	Tolerance Aspiration Concern *
	Therapy Sessions with <20% Aspiration N (%)	Therapy Sessions with VAS ≤ 70 mm N (%)
Case 1	14/14 (100%)	12/12 (100%)
Case 2	11/11 (100%)	6/7 (85.7%)
Case 3	18/18 (100%)	18/18 (100%)
Case 5	20/24 (83.3%)	12/17 (70.6%)
Case 7	8/10 (80.0%)	5/6 (83.3%)
Case 8	11/11 (100%)	11/11 (100%)
Case 10	7/7 (100%)	7/7 (100%)
Criterion	≥80% of the therapy sessions for all participants	≥80% of the therapy sessions for 85% of the participants

* Missing values for the 100 mm visual analogue scale (concern of aspiration).

**Table 5 geriatrics-08-00044-t005:** Intervention usability and acceptability according to the Theoretical Framework of Acceptability (TFA) [43].

	Case 2	Case 3	Case 8	Case 10
**Usefulness** IMI: Mean (SD)	7.00 (0.00)	6.86 (0.38)	6.00 (0.58)	7.00 (0.00)
**TFA domains** [43].				
**Affective attitude** Pleasurable Security	“It was cozy sitting with the foods and liquids”… “looked forward to the meetings”. “I felt secure that the therapist from the hospital was available during course of therapy”.	“It was exciting… inspired to buy some of the given food items”.	“Wonderful with the different taste samples”. “I think it has been good… There hasn’t been anything I didn’t like”.	“I liked getting the good advices”… “and the delicious food”… “I am so grateful for the help”.
**Burden** Appropriate Structure minimizes failure Diary burdensome	Duration, frequency, session length were rated appropriate. “The food diary was hard to remember”.	Duration, frequency, session length were rated appropriate. “The weekly visits helped correcting what I did wrong”. “The food diary was difficult”.	Duration, frequency, session length were rated appropriate. “The food diary was hard to remember”.	Duration, frequency, session length were rated appropriate. “I’ve worked with what I had been taught and then we have talked about it next time”.
**Ethicality** Support	“The therapy was quiet and not stressful… Not disturbing that therapist observed me during therapy- but if strangers … then I’m full”.	“No one has ever talked to me about my swallowing problem… I have been embarrassed by the way I eat… It helps me when there are someone helping me with it”.	“Being provided with good explanations on my problems and how to overcome them”.	“When I commit to something, I do it 100%.”… “How on earth would I have learned it on my own?”… “Being taken seriously is motivating”.
**Intervention coherence** Attentional focus Training materials assist in learning	“I constantly think about what I have learned and that I must bow my head and swallow hard… and then have breaks”.	“To try things out and talk about it”… “I had to be conscious in the beginning”.	“Using the various food and liquid samples were pleasant… help to experience that I could ingest more without pain”.	“Swallowing consciously and in small bites to begin with”. “The different foods and liquids have been very important—how would I have learned it without”.
**Opportunity costs** Flexible schedule	“Nothing has been given up”.	“It has not been a problem”… “We have solved it by looking in my calendar”.	“I have not given anything up to engage in the program… we have planned it”.	“We have solved it… It has been easy to fit the program into my daily life”.
**Perceived effectiveness** Improvements of ingestive skills	“Earlier, I coughed it up again… I have got my life back”.	“I am feeling better”… “It has helped me to eat properly- to drink something- to swallow the food”.	“Opening my mouth more widely when taking in foods and chewing has become better”.	“I can see that it helped, much more than I could imagine”… “It has really helped me. I do not choke that much anymore”.
**Self-efficacy** Capacity	Difficulty levels of training materials were rated appropriate. “I experienced it easy”.	Difficulty levels of training materials were rated appropriate. “In the beginning, I just had to be conscious…, but now it had become a habit to chew and swallow more normal”… “it has become a routine”.	Difficulty levels of training materials were rated appropriate. “There have been no obstacles”… “Chewing more texture without a feeling of danger”… “Feeling progression… then you want more”.	Difficulty levels of training materials were rated appropriate. “The effortful swallow became a routine quickly… although the therapist is not here, I still have to avoid my old way of eating”.

**Table 6 geriatrics-08-00044-t006:** Descriptive information on the putative mediators of change in the ACT-ING program.

	Case 1	Case 2	Case 3	Case 5	Case 7	Case 8	Case 10
**BPNES** (mean (SD))							
Autonomy	N/A	4.50 (1.00)	4.75 (0.50)	N/A	N/A	4.50 (0.58)	5.00 (0.00)
Competency	N/A	4.75 (0.50)	4.75 (0.50)	N/A	N/A	4.50 (1.00)	4.75 (0.50)
Relatedness	N/A	5.00 (0.00)	5.00 (0.00)	N/A	N/A	5.00 (0.00)	5.00 (0.00)
**In-therapy engagement** (median, min–max)							
Task difficulty across sessions/practice complexity	9 (1–16)	10 (1–17)	10 (1–17)	2 (1–8)	2 (1–4)	13 (1–16)	14 (1–17)
No. of task levels across sessions/practice variability	5 (1–8)	4 (1–6)	5 (1–6)	2 (1–3)	5 (2–7)	5 (2–7)	4 (3–6)
No. of sets across sessions/practice distribution	7 (2–11)	5 (2–6)	10 (1–15)	5 (1–11)	5 (1–8)	6 (3–10)	7 (5–8)
Swallow repetitions across sets/practice amount	7 (3–9)	7 (4–12)	8 (5–12)	5 (4–11)	5 (4–9)	8 (4–10)	7 (5–11)
Swallow repetitions across sessions/practice amount	46 (6–74)	30 (10–53)	87 (9–130)	25 (5–61)	25 (4–44)	55 (12–84)	44 (36–67)
**Perceived swallowing capacity** (pre/post-test)							
Liquids: 100 mm VAS	N/A	47/96	26/92	N/A	N/A	95/99	67/100
Foods: 100 mm VAS	N/A	19/66	26/83	N/A	N/A	71/99	19/100

Abbreviations: ACT-ING—ACTivity-based strength- and skill-based training of swallowing to improve INGestion; BPNES—Basic Psychological Needs in Exercise Scale; N/A—not applicable; VAS—visual analog scale.

## Data Availability

Data are only available upon request due to restrictions. The data presented in this study are not publicly available due to Danish legislation. Requests to access the dataset will require an individual inquiry to the Danish Data Protection agency for approval.

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
