# Peer review of "Early Feasibility of an Activity-Based Intervention for Improving Ingestive Functions in Older Adults with Oropharyngeal Dysphagia"

_geriatrics, 2023, doi:10.3390/geriatrics8020044_

Round 1

Reviewer 1 Report

Review report 07-03-2023

A brief summary

Article Early feasibility of an activity-based intervention for improving ingestive functions in older adults with oropharyngeal dysphagia”.

Review

This study is part of a large iterative intervention development process and focused on the early feasibility of the ACT-ING program.

Notably, this study uses a mixed-method, multiple-case study design with quantitative and qualitative data in a convergent approach. This study includes multiple participants with OD engaging in the ACT-ING program, where procedures are replicated for each case representing individual units of analysis. The dosage is 2-3 individual face-to-face therapy sessions per week for up to a maximum of 12 weeks.

According to the authors, the results from this early feasibility study suggest that the 12-week ACT-ING program seems to be safe, well-tolerated, usable, and acceptable in older adults with slight to moderate OD.                                

Rating the Manuscript

The topic of this study is relevant and important, especially for geriatricians, neurologists, and rehabilitation physicians.

The manuscript is clear, relevant for the field and presented in a well-structured manner.

The article contains all of the necessary components, and the methodology is particularly clearly explained.

The figures/tables/schemes are appropriate.

The paper can in principle be accepted after minor corrections (there are some mistakes).

Some questions (rhetorical)

·         The diagnosis of sarcopenia is made based on decreased handgrip strength and lean body mass, using bioelectrical impedance analysis or the densitometry method. In this study, sarcopenia was determined using the SARC-F scale and dynamometry.

·         In other European countries, which have not yet developed complex treatment plans for sarcopenia or OD, community-dwelling older patients are unlikely to be able to receive this level of long-term OD care.

Some remarks:

line 413: extremely – must be Extremely

line 425: givenn – must be given

line 528: pandamic – must be pandemic

line 541: sarcopania – must be sarcopenia

line 542: quanity – must be quantity

line 566: cange – must be change

Reviewer 2 Report

Page 3, line 125, OD verified by the GUSS: GUSS is a type of screening test used to determine whether an instrumental examination should be performed on patients. Relying solely on GUSS to identify or diagnose dysphagia is very risky. Dysphagia should be treated accurately based on the cause of the swallowing disorder identified for each patient. For this purpose, an instrumental test using VFSS or FEES is essential. This should be described as a limitation of this study in the review.

Page 6, line 228: In this study, an intervention called ACTING was applied to improve the swallowing function of patients with aspiration during swallowing. Evidence is needed to establish that aspiration occurs in less than 20% of exercise performance as the success criterion for safety. Even with interventions to improve swallowing function, leaving open the possibility of aspiration occurring during exercise performance may endanger the patient. In fact, in cases 5 and 7 in the 'results' section, the aspiration sign occurred almost 20%. This number is also only estimated through clinical signs, and in reality, aspiration may have occurred more than this.

Page 9, Table 1: Certain exercises are not helpful for all symptoms of dysphagia. Depending on the symptom and cause of dysphagia, the appropriate intervention method differs. Therefore, it is necessary to describe the symptoms of dysphagia for each patient in your study.

Page 10, line 363: Wasn't there a way to confirm whether aspiration actually occurred in the patients? Therefore, it is correct to use the expression ‘clinical sign related to aspiration’ instead of the term ‘aspiration’.

Page 15, line 530: afherenceà adherence?

Page 16, line 566: cangeà change?
